# Neuroprotective Effect of White *Nelumbo nucifera* Gaertn. Petal Tea in Rats Poisoned with Mancozeb

**DOI:** 10.3390/foods12112175

**Published:** 2023-05-28

**Authors:** Ketsarin Intui, Pimchanok Nuchniyom, Jiraporn Laoung-on, Churdsak Jaikang, Ranida Quiggins, Paiwan Sudwan

**Affiliations:** 1Department of Anatomy, Faculty of Medicine, Chiang Mai University, Chiang Mai 50200, Thailand; ketsarin_in@cmu.ac.th (K.I.);; 2Toxicology Section, Department of Forensic Medicine, Faculty of Medicine, Chiang Mai University, Chiang Mai 50200, Thailand

**Keywords:** tea, *Nelumbo nucifera* petal extract, health benefits, mancozeb, nuclear magnetic resonance spectroscopy

## Abstract

*Nelumbo nucifera* Gaertn. (*N. nucifera*) tea is used as food and folk medicine to reduce toxicity in Southeast Asia. Mancozeb (Mz) is used for controlling fungi in agriculture and contains heavy metals. This study aimed to examine the effect of white *N. nucifera* petal tea on cognitive behavior, hippocampus histology, oxidative stress, and amino acid metabolism in rats poisoned with mancozeb. Seventy-two male Wistar rats were divided into nine groups (*n* = 8 in each). Y-maze spontaneous alternation test was used to assess cognitive behavior, and amino acid metabolism was investigated by nuclear magnetic resonance spectroscopy (^1^H-NMR) from blood. There was a significant increase in relative brain weight in the Mz co-administered with the highest dose (2.20 mg/kg bw) of white *N. nucifera* group. The levels of tryptophan, kynurenine, picolinic acid, and serotonin in blood showed a significant decrease in the Mz group and a significant increase in the Mz co-administered with low dose (0.55 mg/kg bw) of white *N. nucifera* group. However, there was no significant difference in cognitive behavior, hippocampus histology, oxidative stress, and corticosterone. This study demonstrated that a low dose of white *N. nucifera* petal tea has a neuroprotective effect against mancozeb.

## 1. Introduction

Tea is the most popular health drink in the world [1]. It is an antioxidant and has been used to treat obesity, cancer, and anxiety, and has been reported to prevent and improve Alzheimer’s disease [1,2,3]. *Nelumbo nucifera* Gaertn. (*N. nucifera*) belongs to the Nymphaeaece family and is known by the common name of sacred lotus, called “Bualuang” in Thai [4,5]. It is mainly distributed and used as food and traditional medicine in Asia and the north of Australia [6]. In Thailand, it has cultural importance as a symbol in Buddhism [7]. Various parts of *N. nucifera* have been reported to have neuroprotective and health benefits. For example, *N. nucifera* rhizomes and flowers improve cognitive function by enhancing neurogenesis and antioxidant activity in rat hippocampi [8,9]. The petals of *N. nucifera* are rich in gallic acid, as well as flavonoids, including luteolin, quercetin, naringenin, isorhamnetin, cyanidin, and delphinidin [10]. The white *N. nucifera* petals have high contents of phenols and tannins, both of which are strong antioxidants and have been shown to decrease the toxicity of mancozeb in Charolais cattle sperm [11,12]. However, there is limited data concerning its effect on the neurological system. Previous studies showed that flavonoids improve learning-memory performance by enhancing blood flow to the brain, as well as improving neuronal protein synthesis to increase the strength of communication between neurons [13,14,15].

The hippocampus is an important brain structure that plays a role in memory and has functional connections with other parts of the central nervous system [16]. It is located in the ventromedial aspect of the temporal horn of the lateral ventricles and has four anatomical subregions: CA1, CA2, CA3, and CA4 [17,18]. The CA3 subregion is important in retention of spatial memory, whereas CA1 is essential in retrieval of memory [19,20,21]. However, the hippocampus is sensitive to pesticides [22]. Pesticides are commonly used in agriculture and can lead to neurological and cognitive disorders [23,24]. Mancozeb is a manganese/zinc compound connected to ethylene-bis-dithiocarbamate and is commonly used in agriculture for controlling fungal infection of vegetables and ornamental plants [25]. It has been reported to be toxic to the neurological system of rodents [26,27]. Previous findings have shown that neuronal degeneration and decreased brain weight, as well as corticosterone hormone increase, occurred in offspring mice after lactating mothers ingested mancozeb [28]. Hormones, such as thyroid hormones, sex hormones, and corticosterone, are important to hippocampal structural organization and functioning [22]. Alterations in hippocampus structure or learning-memory behavior may also be connected to hormonal impairment. Corticosterone stabilizes hippocampal structural plasticity and learning-memory performance in rats [29,30]. It has been shown that stress and excess corticosterone impairs hippocampus neuronal function, as well as learning and memory function [31]. Similarly, previous studies have shown that mancozeb-induced hypercorticosteronaemia leads to decreased neurogenesis and cognitive impairment [30]. Likewise, previous studies showed pyrethroid, another pesticide, decreased kynurenic acid production, which indicates a disturbance of the kynurenine pathway of tryptophan metabolism [32,33].

The kynurenine pathway is a metabolic pathway of tryptophan, which plays a critical role in the pathogenesis of inflammatory and neurodegenerative diseases [34,35]. Low levels of tryptophan were found in elderly patients with neurodegenerative diseases [36]. Previous studies showed low levels of kynurenic acid and picolinic acid are correlated to neurodegenerative diseases such as Parkinson’s disease and Alzheimer’s disease [37,38,39]. Moreover, tryptophan is the precursor of the neurotransmitters serotonin and melatonin through the methoxyindole pathway [40,41]. Serotonin is a biogenic amine and functions as a neurotransmitter correlated to cognitive behavior, sleep, and mood, while melatonin’s main functions are on the sleep cycle and the circadian clock [42,43,44]. It has been reported that, in humans, low levels of serotonin are correlated to depressive and psychiatric diseases [45]. Thus, the aims of the present study were to investigate the effect of white *N. nucifera* petal tea on cognitive function, histology, and antioxidant biomarkers of the hippocampus, as well as corticosterone hormone levels and amino acid metabolism analysis, in Wistar rats poisoned with mancozeb.

## 2. Materials and Methods

### 2.1. Plant Collection and Extraction

White *N. nucifera* petals were collected from Uttaradit Province, Thailand, authenticated at Herbarium, Faculty of Pharmacy, Chiang Mai University, voucher number 023248-2 [11]. The white petals were cleansed in water, steamed, and dried in an oven at 60 °C. The dried petals were ground into fine powder and stored in amber glass bottles before use. The fine powder of white *N. nucifera* petals was extracted with hot water at 75–80 °C (1 mg/mL) for 3–5 min. Then, the solutions were filtered and dried by lyophilization with a 12.5% yield [12] and kept at −20 °C until further use in different doses.

### 2.2. Animal Housing

Seventy-two male Wistar rats of age 6–7 weeks were obtained from Nomura Siam International Co., Ltd., Bangkok, Thailand, and approved by the Ethics Committee of the Faculty of Medicine, Chiang Mai University (32/2021). The animals were acclimatized to laboratory conditions for a week and housed at 2–3 rats per cage in a controlled room at 25 ± 2 °C, with a 12:12 light-dark cycle. Food and water were available ad libitum.

### 2.3. Experimental Design

The male Wistar rats weighed 200–250 g and were divided into nine groups (*n* = 8). Group one received distilled water orally one ml/day. Group two, group three and group four received white *N. nucifera* petals tea at doses of 0.55, 1.10, and 2.20 mg/kg bw, respectively. Group five received olive oil one mL/day as a vehicle group. Group six received mancozeb at dose of 500 mg/kg bw in olive oil. Group seven, group eight, and group nine received mancozeb at a dose of 500 mg/kg bw in olive oil 30 min after receiving white *N. nucifera* petals tea at doses of 0.55, 1.10, and 2.20 mg/kg bw, respectively, for 30 days consecutively. The learning and memory of all rats were assessed by the Y-maze spontaneous alternation test, which is a three-arm maze with equal angles between all arms [46] on the day after the last treatment (day 30). Then, the rats were sacrificed, followed by blood collection for hormonal assays and ^1^H-NMR, and the histology and the antioxidant biomarkers of the hippocampi were studied.

### 2.4. Learning and Memory Behavior Test

The Y-maze spontaneous alternation test was used to assess learning and memory behavior test in this study on the day after the last treatment. The rats were initially placed in the center of the maze and allowed to freely explore for eight minutes. The number of alternations and the total of arm entries were noted by the researcher and video recorded. The percentage of alternation was calculated as (number of alternations/total arm entries − 2)) × 100. The number of alternations in which all three arms were denoted, i.e., ABC, ACB, CAB, CBA, BCA, or BAC (but not, for example, CAC) and was recorded as a measure of short-term memory. The Y-maze was cleaned with 70% alcohol and allowed to dry between use.

### 2.5. Hormonal Essay

After the treatment period, all rats were euthanized with isoflurane, immediately followed by blood collection via the left ventricle of the heart. Whole blood was centrifuged at 3000 rpm for 15 min and plasma levels of corticosterone were measured by the Veterinary Diagnostic Laboratory, Faculty of Veterinary Medicine, Chiang Mai University.

### 2.6. Histological Study

The brains were removed carefully. All brains were weighed, and the right cerebral hemispheres were fixed in a 4% paraformaldehyde solution, dehydrated in alcohol series, cleared in xylene, and then embedded in paraffin wax. The paraffin blocks were cut with a microtome at 3 μm. The sections were mounted on clean glass slides and dried in an oven at 60 °C for 45 min. After deparaffinization in down-grade ethyl alcohol (100%, 95%, and 80%), sections were processed for hematoxylin and eosin staining for observation and photomicrography using a light microscope. The alignment of pyramidal cells was investigated in the middle of the CA1 and CA3 regions, and the vesicular nuclei of pyramidal cells were counted in an area of 1 μm^2^ of both the CA1 and CA3 regions under 40× magnification objective lens with 10× magnification eyepiece lens by using Image J software version 6.0 (Figure 1).

### 2.7. Oxidative Stress Status

#### 2.7.1. Lipid Peroxidase (LPO) Assay

The thiobarbituric acid-reactive species (TBARS) assay was used to assess 100 μL of supernatant of homogenized left cerebral hemisphere. This method followed Laoung-on et al. [11], and then, the mixture solution was measured by a microplate reader at 532 nm (Bio Tek Synergy H4 Hybrid Microplate Reader, BioTek Instruments, Winooski, VT, USA).

#### 2.7.2. Inhibition of Advance Oxidation Protein Products (AOPP) Formation

Briefly, one hundred microliters of supernatant of homogenized left cerebral hemisphere were added to 25 μL of 1.16 M potassium iodine (KI), flowed by 30 min of incubation, after which 10 μL of absolute acetic acid was added. This method followed Laoung-on et al. [11]. The solution was measured by a microplate reader at 340 nm.

#### 2.7.3. Inhibition of Advance Glycation End Products (AGEs) Formation

Two hundred microliters of supernatant of homogenized left cerebral hemisphere were added into 96 well-plate following the method used and reported by Laoung-on et al. [11] and were measured by a microplate reader at excitation wavelength 360 nm and emission wavelength 460 nm.

### 2.8. Amino Acid Metabolism Analysis by ^1^H-NMR

Two mL of blood was collected from each rat, and the plasma samples were stored at −20 °C before analysis.

The samples were thawed. An 800 μL aliquot was added to 2400 μL of chloroform/methanol 1:1 solution, centrifuged (1700 rpm, 4 °C, 15 min), and then the samples were dried by lyophilization. Afterward, the samples were resuspended in deuterium oxide (D2O) and trimethylsilyl propanoic acid (TSP), which was added to provide an internal reference for the chemical shift (0 ppm). Five hundred μL of the solution was transferred to an NMR tube, and the NMR analysis was performed at the Central Science Laboratory, Faculty of Sciences, Chiang Mai University by the nuclear magnetic resonance spectrometer (Bruker NEO 500 MHz, Bruker Daltonics, Billerica, BA, USA).

MestReNova software version 14 was used to analyze the results. The chemical shifts were referred to the internal standard at 0 ppm. Tryptophan was referenced at 7.2 ppm, and kynurenine was denoted at 7.37 ppm, whereas kynurenic acid was denoted at 7.94 ppm and 7.95 ppm was the reference for picolinic acid. Moreover, serotonin and melatonin were found at 3.1 and 7.21 ppm, respectively, as shown in Figure 2.

### 2.9. Statistical Analysis

The data were presented as mean ± standard deviation (SD). Normal distributions were assessed by Kolmogorov–Smirnov test. The average of relative brain weight was statistically analyzed by one-way ANOVA followed by Tukey’s test. The averages of other parameters were analyzed using the Kruskal–Wallis tests. The significance level was set at *p* < 0.05. Statistical analyses were performed using the SPSS program version 22.

## 3. Results and Discussion

### 3.1. Learning and Memory Behavior by Y-Maze Spontaneous Alternation Test

The percentage of alternation of the Y-maze spontaneous alternation test showed no significant difference when compared to the control group at *p* < 0.05 (Figure 3). In contrast, a previous study showed male Wistar rats, after 28 days of exposure to methomyl (a type of pesticide), decreased the percentage of alternations of the Y-maze test due to alterations in synaptic plasticity [23]. Moreover, mancozeb was not reported to induce learning-memory behavior changes in adult rats at a dose of 500 mg/kg bw, but propylthiouracil (another fungicide) has been shown to impair learning and memory in rat offspring at doses as low as 1.6 mg/kg bw by disrupting brain development [47]. Manganese poisoning has been shown to cause behavioral alternations in the Y-maze test in rats [48]. A prior study showed red *N*. *nucifera* petals increased the percentage of alternations in mice with scopolamine-induced memory deficit [49]. In short, different pesticides and fungicides have differing effects on learning and memory behavior, and the current study showed no ill effects from mancozeb, perhaps because the dosage was too low.

### 3.2. Effect of White N. Nucifera Gaertn Petal Tea and Mancozeb on Relative Brain Weight

Relative brain weight showed a significant increase in Mz + NNE 2.20 mg/kg bw group when compared to the olive oil group, as presented in Figure 4. However, the Mz groups tend to increase relative brain weight when compared to the olive oil group (through not significant).

Similarly, a previous studies showed increased relative brain and liver weights in albino rats poisoned with mancozeb at a dose of 500 mg/kg bw for 30 days due to hypertrophy of both tissues [50,51]. Moreover, subchronic treatment with chlorpyrifos-methyl, a type of pesticide, also caused an increase in relative brain weight due to minor histopathological changes in the brain [52]. In contrast, other studies showed decreased relative brain weight in adult male rats after receiving cypermethrin (a pesticide) for 28 days consecutively [53] and a reduction of brain weight of adult rats on mancozeb exposure to a high dose of 1500 mg/kg/day for 360 days [26]. However, evidence shows that changes in relative brain weight are rarely to a neurotoxic level [54]. In short, mancozeb has been shown to induce relative brain weight increase in acute and subchronic periods and decrease relative brain weight at high doses in chronic time scales.

### 3.3. Number of Pyramidal Cells and Histology of Hippocampus

The present study showed no significant difference in number of pyramidal neurons in both CA1 and CA3 regions when compared between groups (Table 1). Both the CA1 and CA3 regions in all groups showed shrinkage and degeneration (darkly stained neurons) of pyramidal neurons, as shown in Figure 5 and Figure 6, and this is likely due to a natural phenomenon seen in pyramidal neurons in the adult neurogenesis of the brain [55].

Adult neurogenesis in the hippocampus plays a role in learning and memory, especially pyramidal neurons, which are large neuronal cells in the hippocampus [56]. Interestingly, a prior study on the effect of pesticides on the hippocampus structure of mice showed significantly decreased density and size of pyramidal cells after pesticide exposure [27]. Moreover, histopathological changes in the hippocampus of rats poisoned with chlorpyrifos and cypermethrin, especially pyramidal cells in CA1 and CA3 regions, were induced by stimulating ChE activity and impairing neuron conductivity [53]. Additionally, a recent study showed that white *N*. *nucifera* petal extract at a dose of 1.10 mg/kg bw [12] had no toxic effect on the hippocampus structure and learning-memory after oral administration for 30 consecutive days [57]. In view of these and the discussion of the cognitive behavior changes above, further research on the connection between histopathological changes and cognitive behaviors would be helpful in the evaluation of mancozeb and white *N*. *nucifera* petal extract effects on brain function.

### 3.4. Corticosterone Level

Corticosterone level was increased in the mancozeb group (though not significantly) when compared between groups. There was no significant difference between all groups when compared between groups, as presented in Figure 7.

Our study agreed with a prior study where mancozeb induced hypercorticosteronaemia in mice after ingestion by lactating mothers for 28 days [30], and is similar to a study in which fipronil pesticide-induced elevated serum corticosterone in adult male rats due to disruption of the hypothalamus-pituitary-adrenal (HPA) axis [58]. Elevated cortisol levels were also observed in *Garra gotyla* fish after exposure to manganese for a week [59]. In contrast, the *N*. *nucifera* flower tea at doses of 10, 100, and 200 mg/kg bw was reported to reduce the level of corticosterone in stress-induced rats [9]. Thus, the difference in these studies, including the present study, may be that the corticosterone effect of mancozeb is correlated to the age of rodents.

### 3.5. Effect of White N. nucifera Gaertn Petal Tea Extract on Oxidative Stress Status

Regarding the antioxidant activity of white *N*. *nucifera* petal extract alone and co-administered with mancozeb, levels of lipid peroxidation (LPO), advanced oxidation protein products (AOPPs), and advanced glycation end products (AGEs) were measured but showed no significant difference (Table 2).

Antioxidants are substances that can prevent, inhibit, and reduce cell damage to DNA, lipids, and proteins of cells due to oxidative stress [60,61]. Free radicals are generated from endogenous sources, such as immune cells, inflammation, mental stress, and also by exogenous sources, including environmental pollutants, smoking, and alcohol [62]. Cell and organelle membranes are made from a double layer of a polar hydrophilic head and a non-polar hydrophobic tail. In most mammalian cell membranes, there are both straight-chained saturated fatty acids and unsaturated fatty acids [63]. Their polyunsaturated fatty acids (PUFAs) are sensitive to oxidative stress, which is known as lipid peroxidation (LPO) [64]. LPO is a process that involves the formation of lipid radicals [65]. In an in vivo study, white *N*. *nucifera* petal extract was shown to reduce LPO of PUFAs in Charolais cattle sperm exposed to mancozeb at low doses of 0.22 and 0.44 μg/mL [12].

Moreover, on the external surface of cell membranes, most of the proteins and some of the lipids are conjugated with short chains of carbohydrates [63]. The most common results of glycoxidation are increased advanced oxidation protein products (AOPPs) and advanced glycation end products (AGEs) [66]. AOPPs are specific markers of protein oxidation, which are made during oxidative stress by the interaction between protein plasma and chlorinated oxidants [63,67], while AGEs are lipids or proteins that become glycated after exposure to sugars [68]. Our present study is similar to an in vivo study of the effect of white *N*. *nucifera* petal extract on Charolais cattle sperm exposed to mancozeb at low doses of 0.22 and 0.44 μg/mL, which showed no significant differences in the AOPPs and AGEs [12]. Furthermore, a prior study showed white *N*. *nucifera* petal extract decreased the levels of oxidative stress markers (2-hydroxybutyric acid, 4-hydroxynonenal, l-tyrosine, pentosidine, and N6-carboxymethyllysine) on rat livers poisoned with mancozeb [50].

### 3.6. Effect of White N. nucifera Gaertn. Petal Tea and Mancozeb on Amino Acid Metabolism by ^1^H-NMR

The results of the amino acid metabolism study of tryptophan, L-kynurenine, kynurenic acid, picolinic acid, serotonin, and melatonin are presented in Figure 8. Tryptophan showed a significant decrease in the Mz group when compared to the olive oil group (Figure 8A). Tryptophan is an essential amino acid in all animals [69]. The main pathway of tryptophan metabolism is the kynurenine pathway, leading to bioactive molecules which are both neurotoxic and neuroprotective and antioxidative [41]. It has been reported that low circulating levels of tryptophan were found in elderly patients with neurodegenerative diseases [70,71]. Tryptophan plays a crucial role in the kynurenine pathway and may serve as a biomarker for monitoring neurodegenerative disease risk. Similarly, a previous study showed reduced tryptophan levels in mice exposed to maneb (a fungicide similar to Mz) [72]. The present study shows that Mz disrupts the level of tryptophan, while white NNE tends to ameliorate this effect.

The Mz group, as well as the co-administered of Mz/NNE groups, all showed a significant decrease in L-kynurenine levels when compared to the control group (Figure 8B). Kynurenic acid is synthesized as tryptophan degrades via the kynurenine pathway [34]. The levels of kynurenic acid showed no significant difference when compared between groups at *p* < 0.05 (Figure 8C). However, the NNE groups showed increased levels of kynurenic acid (though not significant) when compared to the control group, while the level of kynurenic acid in the Mz group was lower than in the olive oil group (though not significant). Likewise, picolinic acid level showed a significant decrease in the Mz group when compared to the olive oil group, while the co-administered Mz with NNE groups at doses of 0.55 and 2.20 mg/kg bw showed a significant increase when compared to the Mz group (Figure 8D). Kynurenic acid and picolinic acid act as neuroprotectives in depression and various neurodegenerative diseases [73]. Previous studies showed depletion of plasma kynurenic acid in mice with multiple sclerosis induced by cuprizone, and in humans, a reduction of plasma picolinic acid was found in those who attempted suicide [74,75]. Depletion of kynurenic acid was also reported in deltamethrin and fenpropathrin poisoning in rat brain slices [32]. Therefore, Mz effects of reducing the level of picolinic acid, and white *N. nucifera* petal tea at low (0.55 mg/kg bw) and high (2.20 mg/kg bw) doses had a beneficial neuroprotective effect against Mz toxicity. Moreover, the picolinic acid showed a significant increase in the NNE 2.20 mg/kg bw group when compared to the control group. Thus, white *N. nucifera* petal tea has a beneficial effect on health promotion.

In addition, the Mz-only group, as well as in the co-administered Mz/NNE groups at doses of 1.10 and 2.20 mg/kg bw, showed significant decreases in serotonin when compared to the olive oil group (Figure 8E). The low-dose NNE appeared to prevent the drop in serotonin due to the Mz. Serotonin plays crucial roles in neural activities, mood, and behavior [45]. Previous studies have shown low levels of serotonin in people with psychosocial stresses such as depression and anxiety [76,77]. Similarly, organophosphate, which is an agricultural pesticide, was associated with anxiety and depression via the serotonin system disruption [78]. Thus, pesticides such as Mz have been repeatedly shown to reduce the levels of serotonin. Additionally, the white *N. nucifera* petal aqueous extracts (NAE) have high contents of total phenolics, total tannins, and antioxidant activity [11], which have been reported to be protective in neurodegenerative diseases and depressive disorders [79,80]. Another study reported *N. nucifera* seeds have antidepressant effects and are closely related to serotonin [81]. In brief, white *N. nucifera* petal tea at a low dose protects serotonin levels after Mz exposure.

Melatonin levels were lower in all the Mz groups, though not at significant levels when compared between groups at *p* < 0.05 (Figure 8F).

## 4. Conclusions

In conclusion, white *N. nucifera* petal tea at doses of 0.55, 1.10, and 2.20 mg/kg bw did not affect relative brain weight, learning-memory behavior, number of pyramidal neurons, histology of the hippocampus, corticosterone hormone level, level of oxidative stress, or products of tryptophan metabolism. However, white *N. nucifera* petal tea at a dose of 2.20 mg/kg bw increased the level of neuroprotective picolinic acid. Mz affected the level of tryptophan, picolinic acid, and serotonin, while white *N. nucifera* petal tea at a dose of 0.55 mg/kg bw increased picolinic acid and serotonin in rats exposed to Mz. From the results, the white *N. nucifera* petal tea at a high dose had beneficial health effects, and at a low dose, it had neuroprotective effects against Mz toxicity.

## Figures and Tables

**Figure 1 foods-12-02175-f001:**
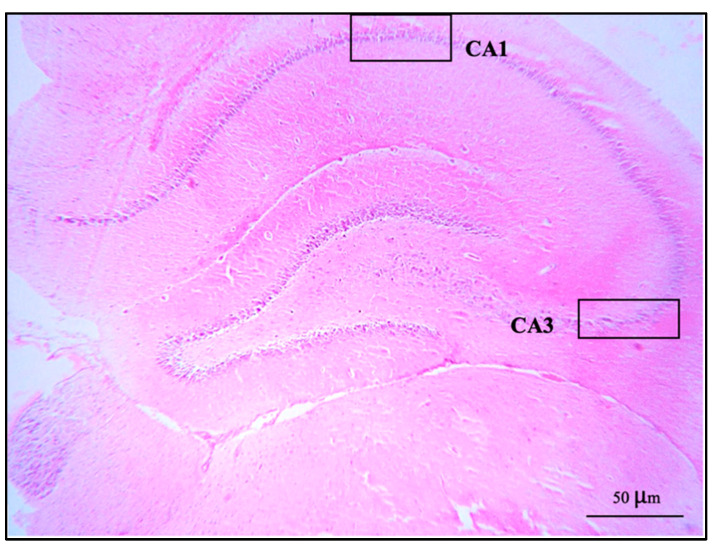
A coronal section of the hippocampus of control group stained with hematoxylin and eosin: CA1 and CA3 regions showed at 10× magnification objective lens with 10× magnification eyepiece lens.

**Figure 2 foods-12-02175-f002:**
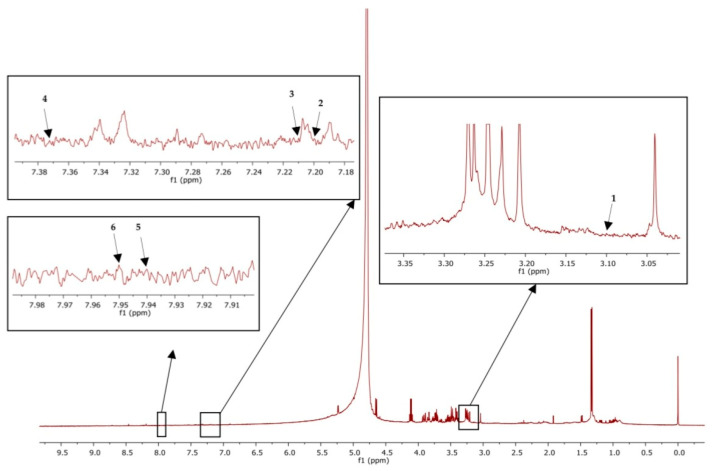
^1^H-NMR of plasma sample showed peak identification of amino acid metabolites: Serotonin (**1**), tryptophan (**2**), melatonin (**3**), kynurenine (**4**), kynurenic acid (**5**), picolinic acid (**6**).

**Figure 3 foods-12-02175-f003:**
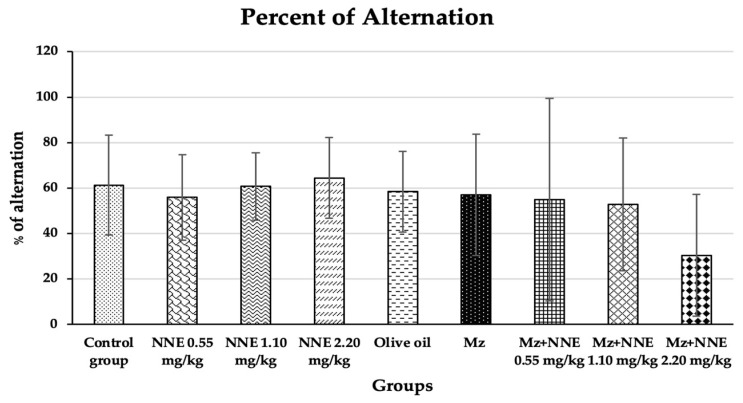
Percentages of alternation of the Y-maze spontaneous alternations test are presented as mean ± standard deviation (SD), *n* = 8. No significant difference in the percentage of alternation at *p* < 0.05 (Percentage of alternations were analyzed by Kruskal–Wallis test).

**Figure 4 foods-12-02175-f004:**
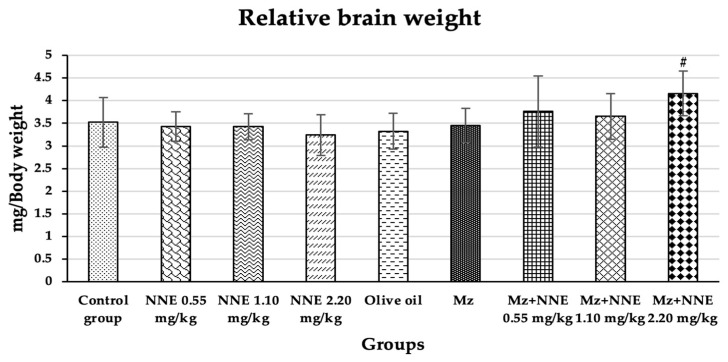
Relative brain weight is presented as mean ± standard deviation (SD), *n* = 8. ^#^
*p* < 0.05 significant difference when compared to the olive oil group (relative brain weights were analyzed by One-way ANOVA followed by Tukey’s test).

**Figure 5 foods-12-02175-f005:**
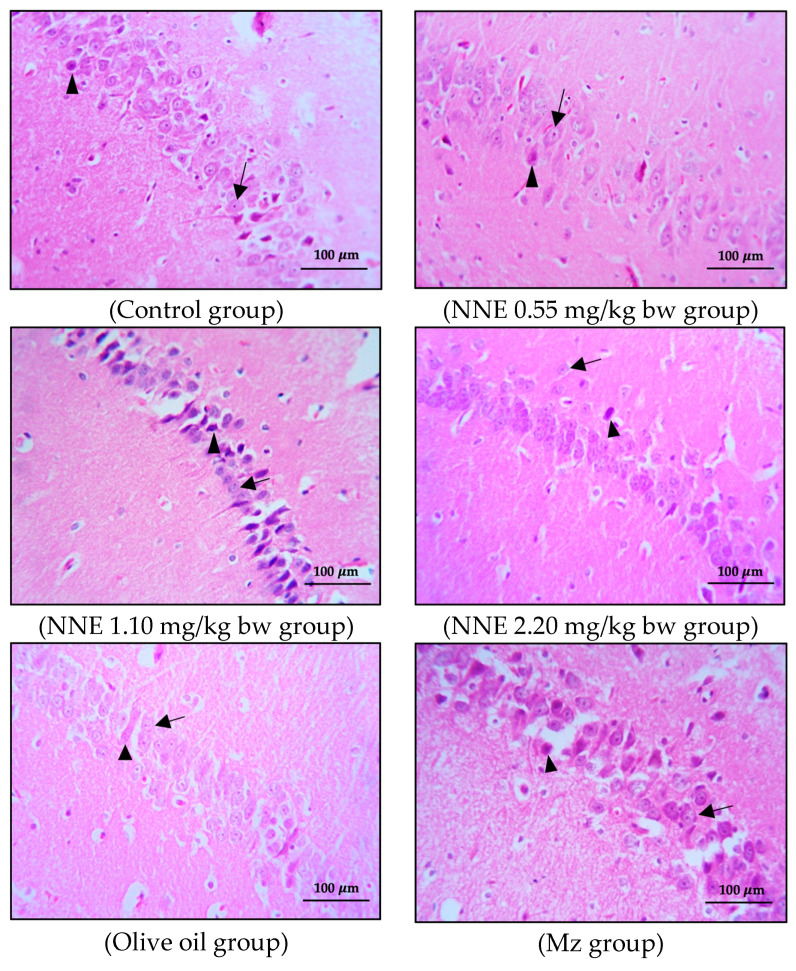
Coronal sections of CA1 region of the hippocampus. Pyramidal neurons were stained with hematoxylin and eosin (H&E). Note the normal cell as the black arrows and the shrinkage and pyknotic cell as the arrowheads.

**Figure 6 foods-12-02175-f006:**
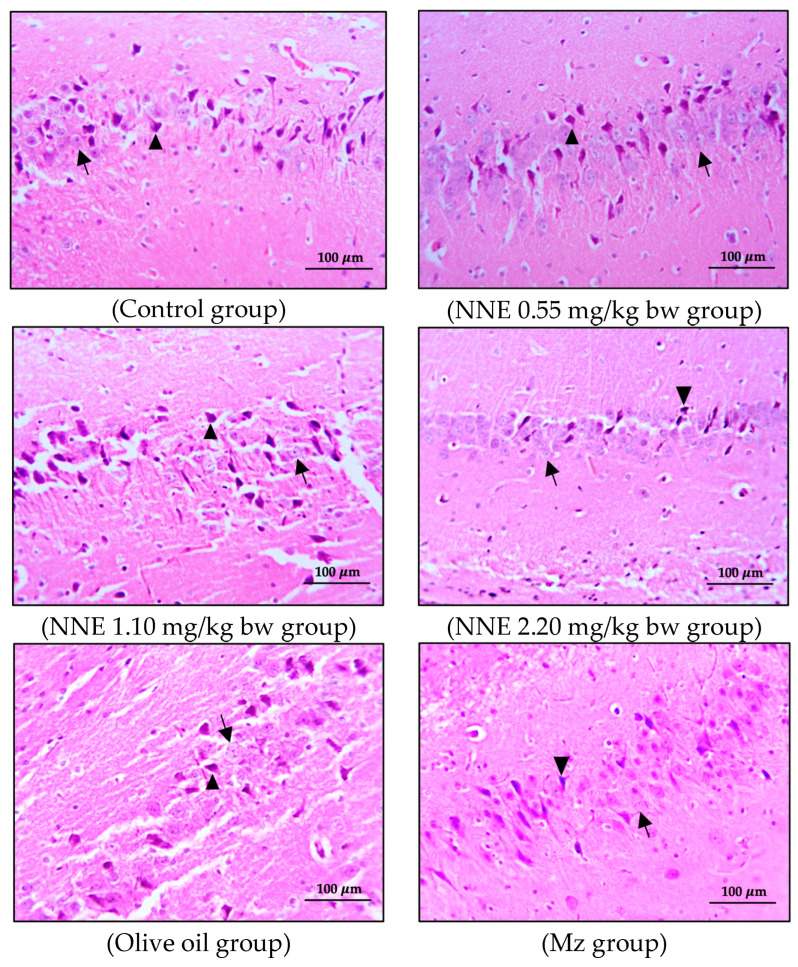
Coronal sections of CA3 region of the hippocampus. Pyramidal neurons were stained with hematoxylin and eosin (H&E). Note the normal cell as the black arrows and the shrinkage and pyknotic cell as the arrowheads.

**Figure 7 foods-12-02175-f007:**
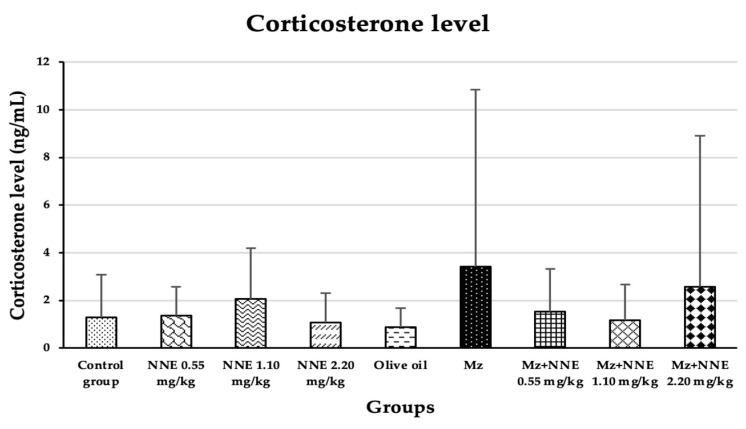
The plasma corticosterone levels are presented as mean ± standard deviation (SD), *n* = 8. No significant difference in the plasma corticosterone hormone level at *p* < 0.05 (Plasma corticosterone level was analyzed by Kruskal–Wallis test).

**Figure 8 foods-12-02175-f008:**
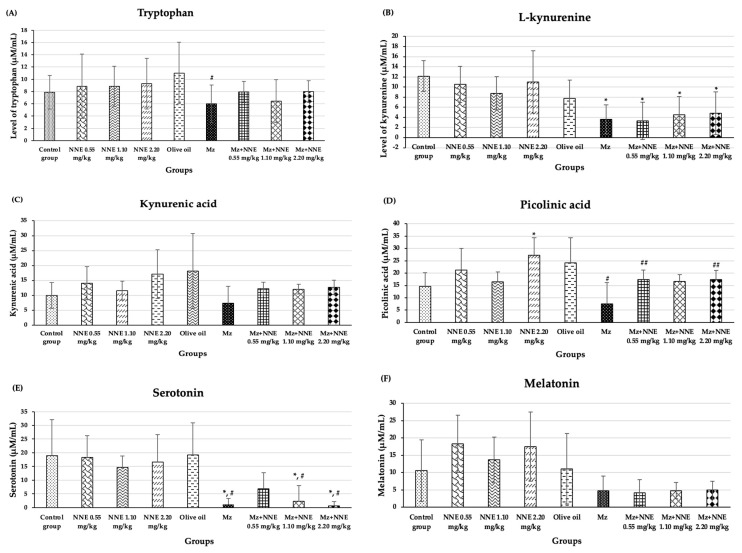
The mean ± SD of products of tryptophan metabolism: Tryptophan (**A**), L-kynurenine (**B**), kynurenic acid (**C**), picolinic acid (**D**), serotonin (**E**), melatonin (**F**). * *p* < 0.05 significant difference when compared to control group. ^#^
*p* < 0.005 significant difference when compared to the olive oil group. ^##^
*p* < 0.05 significant difference when compared to the Mz group.

**Table 1 foods-12-02175-t001:** Number of pyramidal cells in CA1 and CA3 regions of the hippocampus.

Groups	Number of Pyramidal Cells (Cells)	
	CA1 Region	CA3 Region
Control group	24.67± 4.84	15.00 ± 6.63
NNE 0.55 mg/kg	17.14 ± 14.04	9.86 ± 8.40
NNE 1.10 mg/kg	20.50 ± 6.89	14.86 ±13.23
NNE 2.20 mg/kg	22.50 ± 12.18	13.75 ± 9.44
Olive oil	18.83 ± 12.27	16.00 ± 10.41
Mz	24.29 ± 9.38	16.12 ± 10.35
Mz + NNE 0.55 mg/kg	19.63 ±13.21	10.25 ± 6.43
Mz + NNE 1.10 mg/kg	20.14 ± 11.68	11.86 ± 7.10
Mz + NNE 2.20 mg/kg	17.17 ± 16.12	15.67 ± 7.58

All values are presented as mean ± standard deviation (SD), *n* = 8. No significant difference in the number of pyramidal cells at *p* < 0.05 (The number of pyramidal cells in CA1 and CA3 regions was analyzed by Kruskal–Wallis test).

**Table 2 foods-12-02175-t002:** The level of the total oxidative status on the left cerebral hemisphere.

Groups	Oxidative Stress Level		
	**LPO** **(mEq** μ**mol Hydrogen Peroxidase/L)**	**AOPPs** (×10^−3^ mEq μ**mol Chloramine-T/L)**	AGEs (mEq μ**mol Gallic Acid/L)**
Control group	0.41 ± 0.34	2.57 ± 1.10	1.97 ± 1.07
NNE 0.55 mg/kg	0.43 ± 0.26	2.82 ± 0.25	2.44 ± 1.48
NNE 1.10 mg/kg	0.43 ± 0.25	2.47 ± 0.99	1.82 ± 1.45
NNE 2.20 mg/kg	0.41 ± 0.39	2.86 ± 0.22	3.20 ± 2.38
Olive oil	0.50 ± 0.54	2.90 ± 0.21	3.22 ± 2.73
Mz	0.38 ± 0.29	2.85 ± 0.37	2.72 ± 2.37
Mz + NNE 0.55 mg/kg	0.64 ± 0.53	2.94 ± 0.28	3.13 ± 3.35
Mz + NNE 1.10 mg/kg	0.70 ± 0.35	2.92 ± 0.12	2.08 ± 0.81
Mz + NNE 2.20 mg/kg	0.72 ± 0.62	2.98 ± 0.21	3.69 ± 2.68

All values are presented as mean ± standard deviation (SD). No significant difference in the total oxidative status at *p* < 0.05 (The level of LPO, AOPPs, and AGEs were analyzed by Kruskal–Wallis test).

## Data Availability

The authors declare that the data supporting the findings of this study are available within the article.

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
