# Peer review of "Neuroprotective Effect of White Nelumbo nucifera Gaertn. Petal Tea in Rats Poisoned with Mancozeb"

_foods, 2023, doi:10.3390/foods12112175_

Round 1

Reviewer 1 Report

In this manuscript, the neuroprotective effect of white flower petal tea on manganese-zinc poisoning rats was investigated through animal experiments, and the neuroprotective effect of low-dose tea on manganese-zinc poisoning rats was elucidated. The study has some value. However, there are several places in the article that need to be modified and improved, such as the prologue content is long, and it is suggested to simplify;

What was the basis of olive oil selection in the control group?

What are the drugs commonly used clinically for big Xian manganese zinc poisoning? What are the therapeutic advantages of tea over tea?

It is suggested to increase the research on the main components and contents of tea and discuss its neuroprotective mechanism. Pay attention to punctuation.

It is recommended to return for revision and reinspection.

  • General English level.

    • It is better to improve the English writing ability of manuscripts.

    •  

Author Response

  1. What was the basis of olive oil selection in the control group?

Response: Mancozeb is fat-soluble. We used olive oil as vehicle, which followed the previous studies ( Ananthan, 2017, Sardoo, 2018).

  1. What are the drug commonly used clinically for a big Xian manganease zinc poisoning? What are the therapeutic advantage of tea over tea?

Response: Manganese poisoning or manganism leads to neurotoxicity. The mainstay of manganism treatment is sodium para- aminosalicylic acid ( PAS) due to salicylic acid has anti-inflammatory effect, which may contribute to effectiveness of PAS in treatment of neurodegenerative manganism. Previous study showed ascorbic acid can use for treatment of manganese toxicity (Mahmoud, 2018).

Zinc poisoning treatment is chelators to remove excess metals. Drinking milk is also a treatment of zincpoisoning because of calcium and phosphorus in the milk can help bind the excess zinc and prevent the stomach and intestines from absorbing it.
For mancozeb poisoning, it has no evidence of treatment clinically. However, dimercaprol has been show to treatment pesticide poisoning. Vitamin E can also prevent effect of mancozeb (Saddein, 2019).
The advantage of the tea over the drug is the tea have side effects less than the drug and the tea can drink daily for health promotion.

  1. It is suggested to increase the research in the main components and contents of tea and discuss its neuroprotective mechanism.

Response: The sentence was added “ “ Additionally, the white N. nucifera petal aqueous extracts ( NAE) have high contents of total phenolics, total tannins and antioxidant activity [11], which have been reported to be protective in neurodegenerative diseases and depressive disorders [79,80]”.(page 13, line 362-364)

  1. It is better to improve the English writing ability of manuscripts.

Response: This manuscript is edited the English by a native American speaker. .(page 15, line 403-404)

Reviewer 2 Report

Dear Authors,

Congratulations on your interesting research work.

My comments are minor and do not affect the value of the work as a whole. I notice that you use the term Tryptophan in the text and in Figure 8 (L-tryptophan). It would be good to standardise this.

Also, you cite one of your papers (item 11. - Laoung-on, J.; Jaikang, C.; Saenphet, K.; Sudwan, P. Phytochemical screening, antioxidant and sperm viability of Nelumbo 424 nucifera petal extracts. Plants 2021, 10, 1375.), and it might be useful to cite the other paper (Laoung-on, J.; Jaikang, C.; Saenphet, K.; Sudwan, P. Effect of Nelumbo nucifera Petals Extract on Antioxidant Activity and Sperm Quality in Charolais Cattle Sperm Induced by Mancozeb Plants 2022, 11 (5),  637;  https://doi.org/10.3390/plants11050637).
I wish you continued success.

Author Response

  1. 1. I notice that you use the term Tryptophan in the text and in Figure 8 (L-tryptophan). It would be good to standardize this.

Response: We are sorry for the mistake. The wording “ L-tryptophan

” in Figure 8 was changed to “Tryptophan”.(page 14, line 371)

  1. 2. You cite one of your papers (item 11.- Laoung

on, J.; Jaikang, C.; Saenphet, K.; Sudwan, P. Phytochemical screening, antioxidant and sperm viability of Nelumbo nucifera petal extracts. Plants 2021, 10, 1375), and it might be useful to cite the other paper (Laoung-On, J.; Jaikang, C.; Saenphet, K.; Sudwan, P. Effect of Nelumbo nucifera petals extract on antioxidant activity and sperm quality in Charolais cattle sperm induced by mancozeb. Plants (Basel) 2022, 11, doi:10.3390/plants11050637).

Response: I cited “ Laoung- On, J. ; Jaikang, C. ; Saenphet, K. ;

Sudwan, P. Effect of Nelumbo nucifera petals extract on antioxidant activity and sperm quality in Charolais cattle sperm induced by mancozeb. Plants (Basel) 2022, 11, doi: 10. 3390/ plants11050637” followed your suggestion. I added the sentence “The white N. nucifera petals have high contents of phenolic and tannin, both of which are strong antioxidant, and against toxicity of mancozeb in Charolais cattle sperm” (Ref 12). .(page 14, line 371) 

Reviewer 3 Report

The text bellow contains comments on manuscript entitled “Neuroprotective Effect of White Nelumbo nucifera Gaertn. Petal Tea in Rats Poisoned with Mancozeb”.

The manuscript is focused to investigate the effect of white N. nucifera petal tea on the cognitive function, histology, antioxidant properties of the hippocampus in Wistar rats poisoned with mancozeb as well as corticosterone hormone levels and amino acid metabolism analysis. The authors have concluded that a low dose of white N. nucifera petal tea has neuroprotective effect against mancozeb.

To my opinion the manuscript is well written in a sufficient English language level, with accurately structured experimental design. The obtained results fully explain the aim of the study.

I have some minor suggestions for corrections that the authors might take into consideration in case they find them useful.

I think that the introduction section and/or discussion section should contain more examples or related studies if available for the neuroprotective effect of N. nucifera and its major molecules.

Page 3, section 2.1. Please describe for example the amount of dry plant material and water used for extraction.

Minor editing of English language required

Author Response

  1. 1. I think that the introduction section and/ or discussion section should contain more example

or related studies if available for the neuroprotective effect of N.nucifera and its major

molecules.

Response: Three citations were added in the discussion section. “Manganese poisoning has been showed behavioral alternations in the Y-maze task in rats [49].“ was added in the discussion of Learning and memory behavior by Y-maze spontaneous alternation test to mention the effect of manganese poisoning which is component of mancozeb on cognitive behavior. (page 6, line 209-210).

“ Elevated cortisol level was also observed in Garra gotyla fish after being in the tub with exposing of

manganese for a week [59].” Was added in the discussion of Corticosterone level to mention the effect of manganese on cortisol level. (page 11, line 281-282).

Moreover, the sentence “Depletion of kynurenic acid was also reported in deltamethrin and fenpropathrin poisoning in rat brain slice [ 32] . ” Was added in the discussion section in the Effect of white N. nucifera Gaertn. petal tea and mancozeb on amino acid metabolism by 1H-NMR to mention the effect of pesticides on kynurenic acid level (page 13, line 348-349).

  1. 2. Page 3, section 2.1 Please describe for example the amount of dry plant materials and water used

for extraction.

Response: Wording “1 mg/ml” was added in the 2.1 Plant Collection and Extraction section

The amount of dry plant materials and water used for extraction was calculated as following:

The dried material 2,080 g was extracted in the hot water 2,080 ml (1 mg/ml), then the solutions were dried by lyophilization. Finally, There were dried extract 260 g of white N.nucifera Petal.

Thus, the solutions were dried by lyophilization with 12.5 % yield (Page 3 line 91).